# Wakefulness Induced by TAAR1 Partial Agonism in Mice Is Mediated Through Dopaminergic Neurotransmission [note 1]

**DOI:** 10.3390/ijms252111351

**Published:** 2024-10-22

**Authors:** Sunmee Park, Jasmine Heu, Marius C. Hoener, Thomas S. Kilduff

**Affiliations:** 1Center for Neuroscience, Biosciences Division, SRI International, Menlo Park, CA 94025, USA; sunmee.park@sri.com (S.P.); heu@alumni.stanford.edu (J.H.); 2Neuroscience and Rare Diseases Discovery & Translational Area, Roche Innovation Center Basel, F. Hoffmann-La Roche Ltd., CH-4070 Basel, Switzerland; marius.hoener@roche.com

**Keywords:** trace amine-associated receptor 1 (TAAR1), agonists, dopaminergic antagonists, sleep

## Abstract

Trace amine-associated receptor 1 (TAAR1) is a negative regulator of dopamine (DA) release. The partial TAAR1 agonist RO5263397 promotes wakefulness and suppresses NREM and REM sleep in rodents and non-human primates. We tested the hypothesis that the TAAR1-mediated effects on sleep/wake regulation were due, in part, to DA release. Male C57BL6/J mice (*n* = 8) were intraperitoneally administered the D1R antagonist SCH23390, the D2R antagonist eticlopride, a combination of D1R + D2R antagonists, or saline at ZT5.5, followed 30 min later by RO5263397 or vehicle per os. EEG, EMG, subcutaneous temperature, and activity were recorded across the 8 treatments and sleep architecture was analyzed for 6 h post-dosing. As described previously, RO5263397 increased wakefulness and delayed NREM and REM sleep onset. D1, D2, and D1 + D2 pretreatment reduced RO5263397-induced wakefulness for 1–2 h after dosing but only the D1 antagonist significantly reduced the TAAR1-mediated increase in NREM latency. Neither the D1 nor the D2 antagonist affected the TAAR1-mediated suppression of REM sleep. These results suggest that, whereas the TAAR1 effects on wakefulness are mediated, in part, through the D2R, D1R activation plays a role in reversing the TAAR1-mediated increase in NREM sleep latency. In contrast, the TAAR1-mediated suppression of REM sleep appears not to involve D1R or D2R mechanisms.

## 1. Introduction

Trace amines (TAs) such as β-phenylethylamine, m- and p-tyramine, and tryptamine are endogenous amines that are related to the biogenic amine neurotransmitters. Compared to other neurotransmitters, TAs in mammalian brains have much lower concentrations (~1000-fold less) and rapid turnover (<30 s), which makes it difficult to measure them directly [1]. TAs have physiological roles in the mammalian CNS by binding to Trace amine-associated receptors (TAARs), a family of G protein-coupled receptors (GPCRs) that were discovered in 2001 [2,3]. There are six different types of TAARs in humans and nine in rats and mice [4,5]. Unlike other TAARs, which are expressed mostly in the olfactory epithelia in mammals [6], TAAR1 is expressed in the rodent and human brain and has been well-studied compared to other TAARs [7]. TAAR1 is expressed in monoaminergic and limbic system nuclei and is known to modulate dopaminergic (DA) neurons in the ventral tegmental area (VTA), serotonergic neurons in the dorsal raphe nucleus (DRN), and glutamatergic signaling in the prefrontal cortex [7,8,9].

TAAR1 has been extensively studied in relation to psychostimulants over the past 20 years. Multiple studies have shown that the hyperlocomotion caused by elevated DA activity is increased in TAAR1 knockout mice when exposed to psychostimulants such as amphetamine [10,11,12], methamphetamine [10], methylenedioxymethamphetamine (MDMA) [13], and nicotine [14,15]. Conversely, mice that overexpress TAAR1 decrease locomotion after amphetamine administration [16], supporting the concept that TAAR1 is a negative regulator of DA release. TAAR1 full and partial agonists have anti-psychostimulant effects in rodents and non-human primates [17,18]. Recently, ulotaront, a TAAR1 agonist with 5-HT1A agonist activity, entered Phase 3 clinical trials for the treatment of schizophrenia [19,20], raising the prospect of TAAR1 agonism as a clinical intervention for psychiatric disorders involving hyperdopaminergic signaling.

Like other monoamines, DA is known to be involved in sleep/wake regulation since DA neurons in the VTA have been found to be active during wakefulness and rapid eye movement (REM) sleep [21,22,23,24]. We have previously shown that TAAR1 partial agonists promote wakefulness and suppress non-REM (NREM) and REM sleep in mice [25,26], rats [17,18], and non-human primates [27]. Given the role of TAAR1 in the regulation of DA release [7,16], we hypothesized that the wakefulness-promoting effects of TAAR1 partial agonism may be mediated by DA receptor activation. To test this hypothesis, we administered DA receptor antagonists as pretreatments prior to the administration of the TAAR1 partial agonist RO5263397 in mice and monitored sleep/wake control, activity, and subcutaneous body temperature. We confirmed that TAAR1 partial agonism increased Wake Time and reduced both NREM and REM sleep and found that only a combination of D1 and D2 receptor antagonism blocked the TAAR1-mediated increase in wakefulness but that the TAAR1-mediated suppression of REM sleep was independent of DA signaling.

## 2. Results

Appendix A presents example hypnograms for the each of the 8 treatment groups. Since the goal of this study was to determine whether the wakefulness-promoting and sleep-suppressing effects of TAAR1 partial agonism were mediated through DA release, the primary focus of the analyses was on D1 and D2 antagonism of the TAAR1 effects. The experimental design also resulted in the data collection of the D1 and D2 antagonist effects on sleep/wake control. However, since this aspect of sleep/wake control has been addressed elsewhere in the literature [28,29,30,31,32,33], the analyses discussed below focus on TAAR1-DA interactions.

### 2.1. Wakefulness

An RM-ANOVA revealed significant variation in hourly Wake Time due to drug treatment with the TAAR1 agonist RO5263397 (hereinafter RO3397) (Figure 1B–D; F(7, 48) = 3.919, *p* = 0.0019) and a treatment x time interaction (F(35, 240) = 2.693, *p* = 5.2 × 10^−6^). A Tukey’s post hoc test indicated that Saline + RO3397 had wake-promoting effects for the first two hours (ZT6-7) post-dosing compared to Saline + Vehicle (*p* < 0.0001). When Wake Time was analyzed in 3 h bins (Figure 1E–G), Saline + RO3397 increased wakefulness from ZT6 to 8 (*p* < 0.0001) due primarily to more wake bouts (*p* < 0.01; Appendix A) with no significant difference from ZT9 to 11. An RM-ANOVA revealed significant variation in Cumulative Wakefulness over the 6 h recording period (F(35, 240) = 2.882, *p* = 1.1 × 10^−6^; Appendix A); post hoc testing indicated that Saline + RO3397 also increased Cumulative Wakefulness across the entire 6 h compared to Saline + Vehicle (*p* < 0.0001). These results replicate those from our previous studies in mice, rats, and non-human primates [17,18,25,26,27].

Pretreatment with the D1 antagonist SCH23390 followed by RO3397 did not affect Wake Time compared to the Saline + RO3397 combination in either the hourly (Figure 1B) or 3 h ZT6-8 bin (Figure 1E; *p* = 0.13) comparisons. In contrast, pretreatment with the D2 antagonist eticlopride attenuated the increased Wake Time induced by Saline + RO3397 for the first 2 h after dosing (Figure 1C, *p* = 0.0060), although this decrease did not reach significance for the ZT6-8 bin (Figure 1F; *p* = 0.16) even though the mean Wake Bout Duration was reduced by >50% (*p* < 0.0001; Appendix A). However, pretreatment with the D1 + D2 antagonist combination reduced the Saline + RO3397-induced Wake Time increase in both the hourly analysis (Figure 1D; *p* = 0.0075) and the ZT6-8 binned results (Figure 1G; *p* = 0.0086) due to shorter Wake bouts (*p* < 0.0001; Appendix A). When analyzed over the entire 6 h period, the Cumulative Wakefulness induced by the Saline + RO3397 combination was reduced by pretreatment with the D1 antagonist (Appendix A; *p* = 0.0464), the D2 antagonist (Appendix A; *p* = 0.0033), and the D1 + D2 combination (Appendix A; *p* = 0.00163). 

### 2.2. NREM Sleep 

The ANOVA indicated significant variation (F(7, 32.4) = 15.99, *p* < 0.0001) in the latency to NREM sleep produced by different treatments (Figure 2A–C). Post hoc comparisons revealed that Saline + RO3397 significantly increased NREM latency compared to Saline + Vehicle (*p* = 0.0005; Figure 2A–C), the D1 antagonist + Vehicle (*p* = 0.0003; Figure 2A), and the D2 antagonist + Vehicle (*p* = 0.0003; Figure 2B). The Saline + RO3397 increase in NREM latency was blocked by pretreatment with the D1 antagonist + RO3397 (*p* = 0.0004; Figure 2A) but not by the pretreatment with the D2 antagonist + RO3397 (*p* = 0.1264; Figure 2B). Surprisingly, the D1 + D2 antagonist combination had no effect on the RO3397-induced increase in NREM latency (*p* > 0.5, Figure 2C).

An RM-ANOVA indicated significant effects on hourly NREM time across treatments (Figure 2D–F; F(7, 48) = 2.391, *p* = 0.035) and a treatment x time interaction (F(35, 240) = 2.864, *p* < 0.0001). Inverse to the Wake Time results, post hoc tests revealed that Saline + RO3397 significantly reduced NREM sleep for the first 2 h after treatment compared to Saline + Vehicle (Figure 2D–F; *p* = 0.0003). While D1 + RO3397 did not significantly affect NREM time (Figure 2D; *p* = 0.29), post hoc tests revealed that D2 + RO3397 (Figure 2E; *p* = 0.0023) and D1 + D2 + RO3397 (Figure 2F; *p* = 0.0256) significantly increased NREM time compared to the Saline + RO3397 group. Although the two-way ANOVA indicated significant variation in NREM sleep in the 3 h bin analysis from ZT6-8 (F(7, 48) = 4.179, *p* = 0.0012) and Tukey’s post hoc test confirmed that Saline + RO3397 significantly reduced NREM sleep compared to Saline + Vehicle (Figure 2G–I; *p* = 0.0004), only the D1 + D2 + RO3397 combination blocked the Saline + RO3397-induced decrease in NREM sleep (*p* = 0.0246; Figure 2I), due to almost twice as many NREM bouts (*p* < 0.01; Appendix A).

An RM-ANOVA revealed a significant treatment effect on Cumulative NREM Time (Appendix A; F(7, 48) = 5.474, *p* = 0.0001) as well as treatment by time (F(35, 240) = 2.108; *p* = 0.0006). Pretreatment with D1 (Appendix A; *p* = 0.04), D2 (Appendix A; *p* = 0.0007), and D1 + D2 (Appendix A; *p* = 0.005) all attenuated the Saline + RO3397 reduction of NREM sleep. Most of the effect occurred during the initial 3 h (ZT6-8), but there was also some contribution during ZT9-11.

### 2.3. REM Sleep

An ANOVA revealed significant variation in REM sleep latency (Figure 3A–C; F(7, 28.2) = 10.55, *p* < 0.0001), with Tukey’s post hoc test indicating that Saline + RO3397 increased REM latency relative to Saline + Vehicle (*p* = 0.0019). In contrast to NREM sleep, neither pretreatment with the D1 (*p* = 0.43), D2 (*p* > 0.99), nor D1 + D2 antagonist combination (*p* = 0.91) significantly affected the Saline + RO3397 increase in REM sleep latency.

An RM-ANOVA revealed significant effects of treatment (F(7, 48) = 2.458, *p* = 0.031) and treatment x time (F(35, 240) = 1.755, *p* = 0.0079) on hourly REM Sleep Time (Figure 3D–F). Post hoc tests indicated a significant reduction of hourly REM Time by Saline + RO3397, but neither the D1, D2, nor the D1 + D2 combination affected the RO3397-induced decrease in REM Time. Although Figure 3E shows even less REM Time in the D2 + RO3397 group than in the Saline + RO3397 treatment at ZT8, this effect did not reach significance overall. This inhibition of REM sleep was sustained in the 3 h analysis from ZT6 to 8 (Figure 3G–I; *p* = 0.047) without an effect of the DA antagonist treatments. Lastly, although an RM-ANOVA revealed significant effects of treatment (F(7, 48) = 4.091, *p* = 0.0014) and treatment x time (F(35, 240) = 2.188, *p* = 0.0003) on Cumulative REM time, there were no effects of any pre-treatment on RO3397-reduced REM time (Appendix A). 

### 2.4. EEG Spectral Analysis

Appendix A presents the 6 h averaged power spectra for Wake, NREM, and REM sleep, normalized to Saline + Vehicle. Appendix A shows the spectral composition of the conventional bands from ZT6-8 and ZT9-11. As we have observed previously [26], the only significant difference across the various EEG bandwidths is the elevated gamma power in the low (30.2–60.0 Hz) ranges after the Saline + RO3397 treatment.

### 2.5. Subcutaneous Temperature (Tsc) 

Figure 4A–C shows the Tsc for the first 6 h after the second dosing. An RM-ANOVA indicated significant effects of treatment (F(7, 48) = 2.49; *p* = 0.03) and treatment by time (F(35, 240) = 6.18; *p* < 1.10 × 10^−8^) when compared to Saline + Vehicle. Post hoc tests revealed significant treatment effects for D2 + RO3397 (Figure 4B, *p* = 0.017), D1 + D2 + Vehicle (Figure 4C; *p* = 0.013), and D1 + D2 + RO3397 (Figure 4C; *p* = 0.032) but not for Saline + RO3397 (Figure 4A–C; *p* = 0.29) or D1 + RO3397 (Figure 4A; *p* = 0.15). The D2 + 3397 effect was profound across the 6 h recording. As with the D2 + 3397 treatment, the D1 + D2 + Vehicle and D1 + D2 + RO3397 effects were prolonged in duration but not as profound. Post hoc tests for treatment by the time indicated that Saline + RO3397 reduced the Tsc for the first 2 h post-treatment. After the first hour, the D2 + RO3397 (Figure 4B) and D1 + D2 + RO3397 (Figure 4C) treatments appeared to exacerbate the RO3397-induced decline in the Tsc.

### 2.6. Activity 

An RM-ANOVA revealed highly significant effects for both treatment (F(7, 48) = 9.66; *p* = 1.95 × 10^−7^) and the interaction between treatment and time (F(35, 240) = 2.68; *p* = 5.61 × 10^−6^) on locomotor activity (LMA), as defined in the Methods section. When compared to Saline + Vehicle, treatment effects were evident for Saline + RO3397 (Figure 4D–F; *p* = 0.047), D2 + RO3397 (Figure 4E; *p* = 0.0004), and D1 + D2 + RO3397 (Figure 4F; *p* = 0.0001) but not for D1 + RO3397 (Figure 4D; *p* = 0.19). Post hoc tests for treatment by time revealed that all treatments reduced activity relative to Saline + Vehicle for the first post-dosing hour. While LMA recovered after the first hour for the D1 treatment, this inhibition of activity persisted for most of the 6 h period for the 4 treatments that involved the use of the D2 antagonist (Figure 4E,F).

## 3. Discussion

In the present study conducted in mice, the TAAR1 agonist RO5263397 increased wakefulness as well as the latency to NREM and REM sleep, as we have shown previously in three species of mammals [17,18,25,26,27]. Pretreatment with both the D1 and D2 antagonists reduced RO5263397-induced wakefulness during the first 1–2 h after dosing, but only the D1 + D2 antagonist combination attenuated the wake-promoting effect of RO5263397 from ZT6 to 8, mostly by increasing NREM sleep. Although D1 + D2 antagonism blocked the wake-promoting effect of RO5263397, only the D1 antagonist significantly reduced the TAAR1-mediated increase in NREM latency. Surprisingly, neither the D1 nor the D2 antagonist affected TAAR1-mediated suppression of REM sleep. These results suggest that while the TAAR1 effects on wakefulness are mediated, in part, through the D2R, the D1R activation plays a role in reversing the TAAR1-mediated increase in NREM sleep latency. In contrast, the TAAR1-mediated suppression of REM sleep either requires a higher level of receptor occupancy than that achieved in the current study, or it may not involve D1R or D2R mechanisms.

### 3.1. Dopaminergic Effects on Sleep/Wake Control in Rodents

The experimental design utilized here involved the administration of DA antagonists as pretreatments prior to the use of the TAAR1 partial agonist RO5263397. Consequently, it is important to consider the known effects of DA antagonists on sleep/wake control. D1 antagonists such as SCH23390 have been shown to increase both NREM and REM sleep while decreasing wakefulness [28,29,30,31] when administered systemically. In contrast, the effects of D2 receptor antagonists or agonists on sleep depend on the dosage, as D2 receptors are located both pre- and post-synaptically. Eticlopride, a D2 receptor antagonist, activates autoreceptors at low doses, resulting in increased NREM and REM sleep, increased EEG delta power, and reduced locomotor activity. However, high doses of eticlopride activate postsynaptic D2 receptors, leading to arousal and decreased NREM and REM sleep [32,33]. We chose doses of the D1 antagonist SCH23390 (0.25 mg/kg) and the D2 antagonist eticlopride (1 mg/kg) for use in the present study because these doses did not affect sleep/wake amounts in a previous sleep study in mice [34]. Consistent with that study, Table 1 shows that, at those doses, neither SCH23390 + Vehicle, eticlopride + Vehicle, nor the D1 + D2 + Vehicle combination had any effect on Wake (Figure 1E–G), NREM (Figure 2G–I), or REM (Figure 3G–I) sleep time nor on the latency to NREM (Figure 2A–C) or REM (Figure 3A–C) sleep relative to Saline + Vehicle. Thus, despite being “subthreshold” doses for effects on sleep and wake, these doses attenuated the RO5263397-induced wakefulness during the first 1–2 h after dosing, and the D1 + D2 combination had a more sustained effect, supporting our conclusion that the wake-promoting effects of TAAR1 agonists are mediated, at least in part, through D1 and D2 receptor activation.

Nonetheless, D2 + Vehicle did affect sleep architecture by increasing the number of both Wake (*p* < 0.01) and NREM (*p* < 0.05) bouts (Appendix A). The D1 + D2 + Vehicle combination had similar effects, increasing the number of both Wake (*p* < 0.001) and NREM sleep (*p* < 0.05) bouts (Appendix A).

### 3.2. Dopaminergic Antagonism Reduces TAAR1-Mediated Wakefulness

The wake-promoting effects of the partial TAAR1 agonist RO5263397 shown here are consistent with those from our previous studies in mice, rats, and non-human primates [17,18,25,26,27]. TAAR1 is thought to be a negative regulator of DA release [7], and elevated DA levels are normally associated with wakefulness and locomotor activity. Thus, the wake-promoting effects of a TAAR1 partial agonist may seem unexpected but have been interpreted to indicate that RO5263397 antagonizes endogenous TAAR1 tone, increases DA release, and thereby activates DA receptors [17,18]. In contrast, the TAAR1 full agonist RO5256390, which presumably lacks TAAR1 antagonist activity, has no effect on sleep/wake control [18].

At the doses used here, the D1 and D2 antagonists both produced slight, non-significant reductions in the TAAR1-mediated Wake Time increase from ZT6 to 8. However, the D1 + D2 antagonist combination was very effective in reducing this TAAR1-mediated increase in Wake Time (Figure 1G; Table 1). Moreover, when assessed across the entire 6 h recording period, the D1 and D2 antagonists, as well as the D1 + D2 combination, significantly reduced the TAAR1-mediated increase in Cumulative Wake Time (Table 1; Appendix A). Together, these results indicate that the wake-promoting effects of RO3397 are mediated, at least in part, through dopaminergic neurotransmission.

### 3.3. Distinct Dopaminergic Effects on TAAR1-Mediated NREM Sleep Latency vs. NREM Sleep Time

Consistent with the wake-promoting effects of partial TAAR1 agonism shown here and previously, Saline + RO3397 increased the latency to NREM sleep (Figure 2A–C; Table 1). Pretreatment with the D1 antagonist completely blocked this increase in NREM latency (Figure 2A), which is also reflected in NREM Time during ZT6, the first post-dosing hour (Figure 2D). However, neither D2 + RO3397 nor the D1 + D2 + RO3397 combination significantly affected the TAAR1-mediated NREM sleep latency increase. D1 blockade of the TAAR1-mediated increase in NREM latency is consistent with the concept that TAAR1 partial agonists antagonize endogenous TAAR1 tone, which results in DA release, as the D1 antagonist should block post-synaptic D1 receptors and counteract the TAAR1-mediated increase in NREM latency. Only the D1 + D2 + RO3397 combination blocked the TAAR1-mediated reduction in NREM time from ZT6-8 (Figure 2I), but all three treatments mitigated the TAAR1-mediated decrease in Cumulative NREM Sleep Time (Appendix A; Table 1). 

### 3.4. Absence of Dopaminergic Involvement in TAAR1 Effects on REM Sleep or EEG Spectra

In contrast to wakefulness or NREM sleep, DA seems to have no role in the TAAR1-mediated increase in REM latency (Figure 3A–C), the suppression of REM sleep time (Figure 3G–I), or Cumulative REM sleep time (Appendix A), at least at the doses tested. As described previously, RO3397 increased spectral power in the gamma range, but there was no statistical difference after any of the DA treatments, likely due to the large interindividual variation in the Saline + RO3397 treatment group evident in Appendix A.

### 3.5. Tsc and Activity

All of the pharmacological treatments affected the Tsc, although to different magnitudes and on different timecourses. As we have discussed elsewhere [35], while the Tsc can provide a relative measure of Tcore, the Tsc can dissociate from the Tcore and instead reflect activation of heat loss mechanisms through vasodilatation. The Saline + RO3397 treatment transiently reduced the Tsc for the first 2 h after treatment. Curiously, although neither the D2 + RO3397 nor the D1 + D2 + RO3397 treatments affected the Tsc during the first hour after treatment, both D2-related treatments exacerbated the decline in the Tsc evident after Saline + RO3397 and prolonged the effect for several hours. In contrast, the D1 antagonist produced a mild, transient decline in the Tsc.

LMA, as measured by the DSI transmitter, was suppressed by all treatments during the first treatment hour. As with the Tsc, the D1-related effect was transient, whereas the D2-related effects were prolonged. These prolonged effects on the Tsc and LMA are surprising since there were no differences in sleep/wake states evident from ZT9 to 11.

### 3.6. Differential Effects of D1 vs. D2 Antagonism on TAAR1-Mediated Effects

Previous studies have shown functional interactions between D2R and TAAR1 at both pre- and post-synaptic levels [36,37,38,39,40], but no interactions with D1R or modulation of DAT have been found to date [4,8,17,41]. Figure 1B,C and Figure 2D,E, respectively, show distinct differences between the effects of the D1R antagonist SCH23390 and the D2R antagonist eticlopride on TAAR1 agonist-mediated Wake and NREM Sleep. Hourly Wake (Figure 1B) and NREM Time (Figure 2D) indicate that D1 + Vehicle and D1 + RO3397 have similar patterns during ZT6-7 that are significantly different from Saline + RO3397, suggesting that D1 antagonism can mitigate TAAR1 effects. In contrast, D2 + Vehicle and D2 + RO3397 have different patterns from each other; the effect of D2 + RO3397 is more similar to that of Saline + RO3397 from ZT6 to 8 (Figure 1C and Figure 2E), suggesting limited D2 efficacy when paired with RO3397. These results are consistent with those from a study on nucleus accumbens slices in which administration of the D2 antagonist L-741626 blocked the modulation of cocaine-induced DA uptake by the TAAR1 agonist RO5256390 [36]. Consistent with the efficacy of D1 + D2 + RO3397 treatment to block the TAAR1-mediated increase in Wake (Figure 1G) and decrease in NREM (Figure 2I) from ZT6 to 8, the Wake and NREM patterns produced by this combination are also distinct. Perhaps due to the half-life of these compounds, the significant hourly results from these three conditions are only partially reflected in the 3 h bins as only the D1 + D2 + RO3397 combination blocks the TAAR1-mediated increase in Wake and reduction in NREM sleep. Consequently, Table 1 summarizes the results for ZT6-7 as well as for the 3 h ZT6-8 bin.

The D1 and D2 antagonists also differed in their effects on the TAAR1-mediated increase in NREM latency: while the D1 antagonist blocked this increase, the D2 antagonist had no effect. Whether this difference reflects a differential involvement of D1 vs. D2 receptors in facilitating sleep onset or simply different pharmacokinetic/pharmacodynamic properties of the compounds tested is unclear at this time, but this difference is related to the differential amounts of wakefulness in the first hour after dosing (Figure 1B vs. Figure 1C). In contrast to NREM sleep, the TAAR1-mediated increase in REM sleep latency was unaffected by antagonism of either the D1R or D2R.

Overall, the blockade of dopaminergic neurotransmission using D1 and D2 antagonists affected wakefulness and NREM sleep but had no effect on REM sleep, as summarized in Table 1. The differential effects of D1 + RO3397 and D2 + RO3397 on Wake and NREM time are particularly evident in the initial post-dosing hours (ZT6-7), where D2 + RO3397 appears to alter the RO3397-induced increase in Wake and decrease in NREM. The Tsc and LMA show similar patterns to wake, and NREM in that D1 + RO3397 produces an hourly pattern similar to Sal + RO3397 (Figure 4B,E), whereas D2 + RO3397 exacerbated the Tsc decline produced by Sal + RO3397 (Figure 4B). Due to the different half-lives of the D1 and D2 antagonists and RO3397, direct comparison between treatments is limited. Nonetheless, these results indicate complex interactions between DA neurotransmission and TAAR1 partial agonism on physiology and behavioral states. 

### 3.7. Limitations of the Present Study

The experimental design utilized here involved the administration of a single dose of DA antagonists as pretreatments prior to the use of the TAAR1 partial agonist RO5263397. As summarized above, the DA system has profound effects on sleep/wake control in both rodents and humans. However, the doses of the D1 and D2 antagonists we chose had previously been shown to be without effect on sleep/wake control in a study from a well-regarded laboratory [34] and, as summarized above, these doses, when combined with the vehicle, were without effect in our study as well. Nonetheless, many of the components of the DA system are known to be rhythmic at either the mRNA or protein level, and it is conceivable that testing at another time of day might yield different results.

Secondly, we used pharmacological pre-treatment with D1 and D2 antagonists rather than evaluating the effects of the TAAR1 agonists in D1 and/or D2 receptor knockout (KO) mice. Further studies might employ such animal models, although such approaches would require sleep/wake phenotyping of the receptor KO strain.

Thirdly, our approach involved systemic drug administration to establish the principle that the DA system is involved in TAAR1 agonist-mediated effects on wakefulness. Future studies may be directed towards elucidating the underlying circuits using approaches such as local drug injection or more sophisticated tools such as cell-specific knockouts, fiber photometry, or GRAB sensors [42,43] to measure the activity of the DA system.

## 4. Materials and Methods

### 4.1. Animals

Adult (>9 weeks) male C57BL6/J mice (*n* = 8) were used; all mice were >23 g at the time of surgery. Mice were maintained on a 12:12 light/dark cycle with food and water ad libitum. Room temperature (23 ± 2 °C), humidity (50 ± 20% relative humidity), and lighting conditions were monitored continuously (Rees Scientific, Trenton, NJ, USA). All studies were conducted following the Guide for Care and Use of Laboratory Animals and were approved by the Institutional Animal Care and Use Committee (IACUC) at SRI International. 

### 4.2. Surgical Procedures

Mice were implanted with sterilized F20-EET wireless transmitters (DSI, Inc., St. Paul, MN, USA) subcutaneously under their left dorsum to record electroencephalogram (EEG), electromyogram (EMG), locomotor activity (LMA), and subcutaneous body temperature (Tsc). During surgery, mice were anesthetized with 1.5–2% isoflurane, and EMG and EEG leads were routed subcutaneously. EEG leads were inserted through the intracranial burr holes, with one lead placed over the hippocampal area (−2 mm A/P from bregma, +2 mm M/L) and the ground lead over the cerebellum (−1 mm A/P from lambda, +2 mm M/L). After placement of the EEG leads, dental cement was applied to the skull to affix the wires. EMG leads were sutured to the right nuchal muscle. An analgesia cocktail of meloxicam (5 mg/kg, s.c.) and buprenorphine (0.05 mg/kg, s.c.) was administered for two days post-surgery. Meloxicam was administered for an additional day.

### 4.3. EEG, EMG, LMA and Tsc Recording

Three weeks after surgery, mice were acclimated to handling and oral gavage dosing at least seven days before data collection. During each recording session, the wireless transmitters were activated at ZT23 (Day 0) and turned off at ZT24 the following day (Day 1). Physiological signals collected from transmitters were thus continuously recorded for 25 h using Ponemah (DSI, Inc., St. Paul, MN, USA) and subsequently analyzed. Due to signal attenuation in the F20-EET transmitters, the cutoff for EEG spectral analysis was 60 Hz.

### 4.4. Experimental Protocols

As illustrated in Figure 1A, all mice received experimental treatments at both ZT5.5 and 30 min later at ZT6; the experimenter was blinded to the dosing condition. The D1 receptor antagonist SCH23390 (hereafter D1; 0.25 mg/kg), the D2 antagonist eticlopride (hereafter D2; 1 mg/kg), a combination of D1 + D2, or saline were administered intraperitoneally (i.p.) at ZT5.5. The doses of the D1 antagonist SCH23390 and the D2 antagonist eticlopride were selected as these doses did not affect sleep/wake amounts in a previous sleep study conducted in mice [34]. After 30 min, the TAAR1 agonist RO5263397 (“3397”; 1 mg/kg in the vehicle) or vehicle (10% DMSO) was administered at ZT6 per os (p.o.). This dose of 3397 and time of day was chosen because our previous study had shown a robust wake-promoting effect of 3397 when administered at ZT6 [26]. Although the T½ of 3397 was determined to be 6.0 h when mice were dosed at 4 mg/kg, p.o. [18], both the wake-promoting and REM sleep-suppressing effects of 3397 dissipated within 3 h when mice were dosed at 1 mg/kg, p.o. [26]. A repeated measures design was utilized, and all mice received all treatment combinations, and drug treatments were administered in a balanced order. To ensure adequate wash-out between each treatment, at least 72 h elapsed between dosings.

### 4.5. Drugs

SCH23390, eticlopride, and RO5263397 (Sigma Aldrich, St. Louis, MO, USA) were prepared as fresh solutions daily, with sonication for 15 min, and then diluted serially using deionized water (DI H2O) or 10% DMSO (Sigma Aldrich, St. Louis, MO, USA) in DI H2O as the vehicle. Drugs were administered at a volume of 20 mL/kg.

### 4.6. Data and Statistical Analysis

EEG and EMG recordings, collected for 6 h after the second dosing (ZT6-ZT12), were scored in 10 s epochs as wakefulness, NREM, or REM sleep using Neuroscore (DSI Inc., Chanhassen, MN, USA) by expert scorers blinded to drug treatments as described previously [25,26,44]. The resultant data were analyzed using custom MATLAB (Mathworks, MA, USA) scripts and Prism 9 (Graphpad, CA, USA). Dependent variables measured included hourly percent time in Wake, NREM, and REM sleep, cumulative time in Wake, NREM, and REM, measures of sleep/wake consolidation (mean hourly bout duration and the number of bouts of Wake, NREM, and REM), EEG spectral analysis of the EEG for each state, and the effects on the Tsc and LMA. EEG spectra were analyzed in 0.122 Hz bins and then grouped into the standard frequency bands rounded to the nearest Hz (delta: 0.5–4 Hz, theta: 4–9 Hz, alpha: 9–12 Hz, beta: 12–30 Hz, and low gamma: 30–60 Hz). For each mouse, power was normalized to the average power per bin during the 6 h Saline + Vehicle recording. Due to poor signal quality on one of the dosing days, all recordings for that day were excluded from the analysis; consequently, 7 of the 8 possible treatment combinations were analyzed for each mouse. A two-way repeated measures ANOVA (RM-ANOVA), with drug treatment and time as factors followed by Dunnett’s post hoc test, or a one-way ANOVA followed by Tukey’s multiple comparison test, was used to assess statistical significance. The statistical tests used for each figure panel are described in Appendix A.

## 5. Conclusions

TAAR1 has been shown to play an important role in modulating monoaminergic neurotransmission, particularly in relation to psychostimulants and addiction, and TAAR1 agonists are increasingly recognized as a novel mechanism for treatment of psychiatric disorders [20,45,46,47]. Similar to RO5263397, ulotaront, a TAAR1 agonist with 5HT1A agonist activity, has been shown to have wake-promoting and REM sleep-suppressing activity in both preclinical [48] and clinical [49] studies. Ulotaront has also recently been shown to reduce REM sleep without atonia in human subjects [19]. Since ulotaront (previously, SEP-363856) inhibits the firing of VTA neurons [48], it likely also affects DA release as part of its mechanism of action. However, the mechanisms underlying TAAR1 partial agonist effects to promote wakefulness and suppress REM sleep have not been extensively studied. The results of the current study indicate that D2 antagonists can attenuate or block TAAR1-mediated wakefulness without affecting NREM sleep latency, whereas D1 antagonists can counteract the delayed NREM latency produced by TAAR1 agonism. Whether these differential effects reflect differential D1 vs. D2 involvement in sleep onset vs. wake maintenance or a differential time course of action of the particular compounds studied is unclear at this time. Nonetheless, it is noteworthy that neither D1 nor D2 antagonists affected the TAAR1 suppression of REM sleep, particularly since many psychoactive compounds, such as antidepressants, suppress REM sleep.

## Figures and Tables

**Figure 1 ijms-25-11351-f001:**
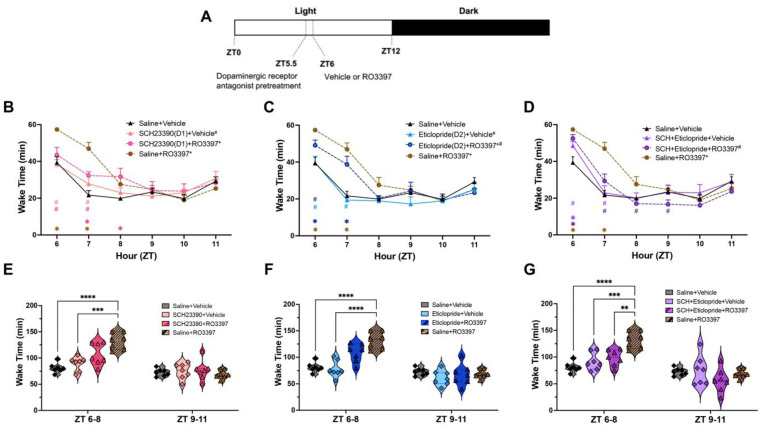
Schematic illustrating the experimental design and Wake Time for 6 h after the second dosing. (**A**). Male C57BL/6 mice received a dopaminergic antagonist i.p. at ZT5.5 followed 30 min later by either p.o. RO3397 or saline at ZT6. (**B**–**G**). For ease of visualization, data are split into three subgroups in which the results from the negative (Sal + Veh) and positive (Sal + RO3397) control treatments are repeated in each graph. Hourly Wake Time (mean + SEM) for the first 6 h after the second dosing with (**B**) D1 antagonist SCH23390 +Veh or +RO3397, (**C**) D2 antagonist eticlopride +Veh or +RO3397, and (**D**) D1 + D2 antagonist + Veh or +RO3397. Colored symbols indicate statistical significance for that hour compared to Sal + Veh (*) or Sal + RO3397 (#) based on RM-ANOVA. Panels (**E**–**G**) present Wake Time (mean + SEM) summed in 3 h bins (ZT6-8 and ZT9-11) for each treatment. *, # *p* < 0.05; **, *p* < 0.01; ***, *p* < 0.005; ****, *p* < 0.001.

**Figure 2 ijms-25-11351-f002:**
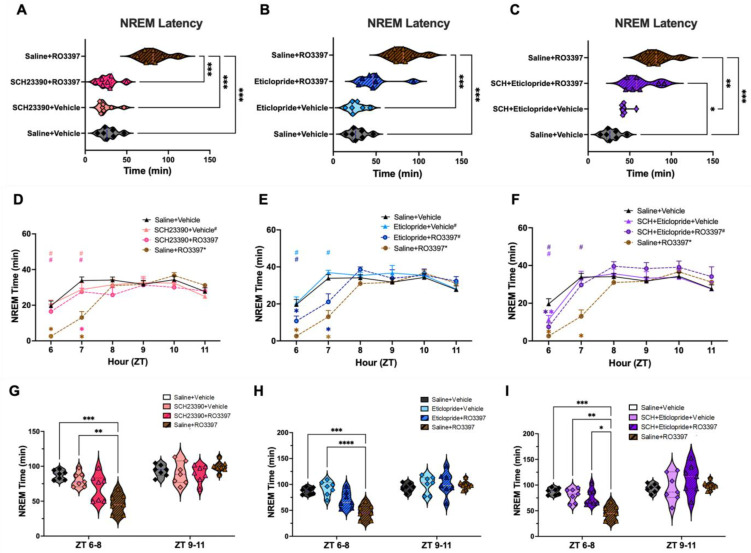
Latency to NREM sleep and NREM time for the first 6 h after the second dosing. For ease of visualization, data are split into three subgroups in which the results from the negative (Sal + Veh) and positive (Sal + RO3397) control treatments are repeated in each graph. (**A**–**C**). Latency to NREM sleep (mean + SEM). (**D**–**F**). Hourly NREM time (mean + SEM) for the first 6 h after the second dosing with (**D**) D1 antagonist + Veh or +RO3397, (**E**) D2 antagonist + Veh or +RO3397, and (**F**) D1 + D2 antagonist + Veh or +RO3397. Colored symbols indicate statistical significance for that hour compared to Sal + Veh (*) or Sal + RO3397 (#) based on RM-ANOVA. (**G**–**I**). NREM Time (mean + SEM) summed in 3 h bins (ZT6-8, ZT9-11) for each treatment. *, *p* < 0.05; **, *p* < 0.01; ***, *p* < 0.005; ****, *p* < 0.001.

**Figure 3 ijms-25-11351-f003:**
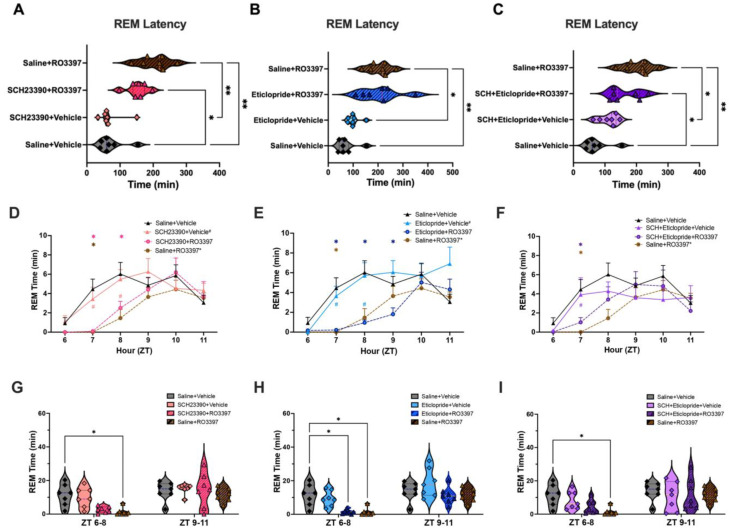
Latency to REM sleep and REM time for the first 6 h after the second dosing. For ease of visualization, data are split into three subgroups in which the results from the negative (Sal + Veh) and positive (Sal + RO3397) control treatments are repeated in each graph. (**A**–**C**). Latency to REM sleep (mean + SEM). (**D**–**F**). Hourly REM time (mean + SEM) for the first 6 h after the second dosing with (**D**) D1 antagonist + Veh or +RO3397, (**E**) D2 antagonist + Veh or +RO3397, and (**F**) D1 + D2 antagonist + Veh or +RO3397. Colored symbols indicate statistical significance for that hour compared to Sal + Veh (*) or Sal + RO3397 (#) based on RM-ANOVA. (**G**–**I**). REM time (mean + SEM) summed in 3 h bins (ZT6-8, ZT9-11) for each treatment. *, # *p* < 0.05; **, *p* < 0.01.

**Figure 4 ijms-25-11351-f004:**
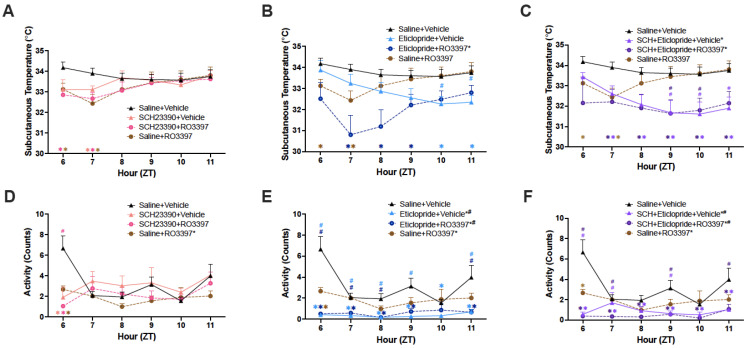
Subcutaneous body temperature and activity for the first 6 h after the second dosing of male C57BL/6 mice that received a dopaminergic antagonist (D1, D2, or D1+D2) i.p. at ZT5.5 followed 30 min later by either p.o. RO3397 or saline at ZT6. (**A**–**C**). Body temperature in mice treated with a D1 antagonist (**A**), D2 antagonist (**B**), or D1+D2 antagonists (**C**) followed by RO3397 or saline. (**D–F**). Activity levels in mice treated with a D1 antagonist (**D**), D2 antagonist (**E**), or D1+D2 antagonists (**F**) followed by RO3397 or saline. Colored symbols indicate statistical significance (*p* < 0.05) for that hour compared to Sal + Veh (*) or Sal + RO3397 (#) based on RM-ANOVA. *, # *p* < 0.05.

**Table 1 ijms-25-11351-t001:** D1 and D2 antagonist effects on physiological parameters when followed by vehicle or by the TAAR1 partial agonist RO5263397.

Parameter	Change Relative to Saline + Vehicle				Change Relative to Saline + RO5263397		
	D1 Antagonist + Vehicle	D2 Antagonist + Vehicle	D1+D2 Antagonists + Vehicle	Saline + RO5263397	D1 Antagonist + RO5263397	D2 Antagonist + RO5263397	D1+D2 Antagonists + RO5263397
Wake Time (ZT6-7)	No effect	No effect	No effect	Increased	No effect	Attenuated increase	Reduced increase
Wake Time (ZT6-8)	No effect	No effect	No effect	Increased	No effect	Not significant	Reduced increase
Cumulative Wake Time (ZT6-11)	Increased	No effect	No effect	Increased	Reduced increase	Reduced increase	Reduced increase
Latency to NREM Sleep	No effect	No effect	No effect	Increased	Blocked increase	No effect	No effect
NREM Sleep Time (ZT6-7)	No effect	No effect	No effect	Decreased	No effect	Increased	Increased
NREM Sleep Time (ZT6-8)	No effect	No effect	No effect	Decreased	No effect	Not significant	Increased
Cumulative NREM Time (ZT6-11)	No effect	No effect	No effect	Decreased	Attenuated decrease	Attenuated decrease	Blocked decrease
Latency to REM Sleep	No effect	No effect	No effect	Increased	No effect	No effect	No effect
REM Sleep Time (ZT6-8)	No effect	No effect	No effect	Decreased	No effect	No effect	No effect
Cumulative REM Time (ZT6-11)	No effect	No effect	No effect	Decreased	No effect	No effect	No effect
Subcutaneous Body Temperature	Mild ↓: ZT7	Delayed ↓: ZT10-11	Prolonged ↓: ZT7-11	ZT6-7 only	Mild ↓: ZT6-7	Profound ↓↓: ZT7-9	Prolonged ↓: ZT7-11
Activity	Reduced at ZT6 only	Reduced throughout	Reduced throughout	Reduced at ZT6 only	Reduced at ZT6 only	Reduced throughout	Reduced throughout

## Data Availability

The original data underlying this study will be deposited and made openly available at the National Sleep Research Resource (www.sleepdata.org) after 1 January 2025.

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
