# Peer review of "Wakefulness Induced by TAAR1 Partial Agonism in Mice Is Mediated Through Dopaminergic Neurotransmissionâ€"

_ijms, 2024, doi:10.3390/ijms252111351_

Round 1
Reviewer 1 Report
Comments and Suggestions for Authors
In this manuscript by Park et al., the authors analyzed interactions between TAAR1 partial agonism and dopaminergic neurotransmission on sleep/wake timing and architecture, locomotor activity and subcutaneous temperature in mice. They found that pretreatment with the combination of D1+D2 antagonists attenuated TAAR1-mediated wake promoting effect. Whereas only D1 antagonist significantly reduced increase in NREM latency, neither the D1 nor D2 antagonist influenced REM suppression induced by treatment with the TAAR1 agonist. Furthermore, these pharmacological treatments transiently and variably affected mice locomotor activity and their subcutaneous body temperature.
In general, the manuscript is well-written, experiments properly designed and executed, and obtained results corroborate conclusions. However, to communicate the data with multiple treatment combinations such as this study, in a more effective and easy-to-digest manner, the paper needs minor adjustments and corrections regarding presentation of results.
1. Please express data using the Super plots graph style (see doi.org/10.1083/jcb.202001064) and merge multiple graphs of the same measurement in one (e.g. Fig.1B-D; Fig.2A-C, etc). This will be particularly helpful for better simultaneous visualization of the effect size of different drug treatments and prevent unnecessary repetitions of the reference treatments (saline+vehicle, saline+RO3397) in multiple graphs as is now. Also, it will enable transparent display of distribution and reproducibility of data.
2. Please include representative hypnograms for each treatment group as a supplemental figure.
3. Denote statistical significance where necessary for EEG power spectrum analyses in Fig. S4.
Reviewer 2 Report
Comments and Suggestions for Authors
In the article entitled: “Wakefulness Induced by TAAR1 Partial Agonism in Mice is Mediated Through Dopaminergic Neurotransmission”, the authors study the involvement of D2 and D1 receptors in the effects of trace amine-associated receptor TAAR1 activation on wakefulness, NREM sleep latency and suppression of REM sleep. The relevance of this study is highlighted by the use of a well characterized in vivo model, and the results shown here contribute to elucidate the mechanisms by which TAs and their receptor modulate DA neurotransmission.
Although the article is clear and well written there are some points that I would like to highlight:
Line 74, in the beginning of the paragraph specify the name of the drug and the effects, i.e. “due to treatment with the TAAR1 agonist RO5263397”.
Line 76, RO3397, has not been abbreviated before.
Line 105, specify the name of the D receptors antagonists; also, in in graph B legend, add (D1)
Line 194: explain better how actvity was measured and quantified. In the graph ”counts” are represented: counts of what?
Line 196: lma has not been abbreviated before.
Line 287: this increase means something? Try to expand the discussion
Comments on the Quality of English LanguageMinor editing of English language required
